# Intermediate Crack Debonding of Externally Bonded FRP Reinforcement—Comparison of Methods

**DOI:** 10.3390/ma15207390

**Published:** 2022-10-21

**Authors:** Paweł Tworzewski, Jeffrey K. Alexy, Robert W. Barnes

**Affiliations:** 1Department of Strength of Materials, Concrete and Bridge Structures, Kielce University of Technology, aleja Tysiąclecia Państwa Polskiego 7, 25-314 Kielce, Poland; 2Brown+Kubican Structural Engineers, 2224 Young Dr, Lexington, KY 40505, USA; 3Department of Civil and Environmental Engineering, 238 Harbert Engineering Center, Auburn University, Auburn, AL 36849, USA

**Keywords:** debonding, FRP, RC beams, strengthening, failure modes, EBR

## Abstract

Many researchers around the world have made extensive efforts to study the phenomenon of fiber-reinforced polymer (FRP) debonding. Based on these efforts, code provisions and various models have been proposed for predicting intermediate crack (IC) debonding failure. The paper presents a comparison of seven selected models: *fib* bulletin 14 approach, Teng et al. model, Lu model, Seracino et al. model, Said and Wu model, Elsanadedy et al. model and ACI 440. The accuracy of each model was evaluated based on the test results of 58 flexural specimens with IC debonding failures of externally bonded (EB), carbon FRP plates or sheets found in the existing literature. The experimental database was prepared to include a wide range of parameters affecting the issue under consideration. A comparison of the measured and predicted load capacity values was made to evaluate the prediction accuracy of the considered models. The analysis included the limitation of the load capacity estimated based on IC debonding models as well as concrete crushing and FRP rupture types of failure. The results indicate that the latest models proposed for direct implementation in design guidelines—the Said and Wu model and the Elsanadedy et al. model—offer the best accuracy in predicting the load capacity. In contrast, the *fib* bulletin 14 approach shows a wide dispersion of predictions and a large proportion of highly overestimated results.

## 1. Introduction

External strengthening is the most popular method used to increase the load-bearing capacity of reinforced concrete (RC) members. Externally bonded (EB) systems include steel strengthening or composite strengthening. Fiber-reinforced polymer (FRP) composites are commonly used due to their strength parameters, corrosion resistance and light weight; the possibility of using any lengths without joints; and the ease of transport and application [1,2,3]. 

Several failure modes of RC beams strengthened with an externally bonded, tension-face FRP sheet can be distinguished. The first mechanism is intermediate crack (IC) debonding. The second mechanism consists of the rupture (R) of the composite reinforcement in the middle of the element. Other mechanisms include shear; crushing of compressive concrete (CC); and loss of anchorage at composite ends, which includes plate end (PE) debonding, concrete cover separation (CCS) and anchorage (A) failure [3,4].

Reinforced concrete members strengthened with bonded plates usually do not achieve flexural failure (concrete crushing or rupture of bonded plate). More typical are PE or IC debonding failures. Anticipating such failure is important for safe and efficient strengthening design. PE debonding failure can be eliminated with the use of proper anchorage (e.g., by using integrated FRP composite anchors [5] or FRP U-jackets). In the case of an IC debonding fracture that takes place in the adhesive layer, it can be eliminated by proper concrete surface preparation and application of good-quality adhesives [6]. However, when the IC debonding fracture occurs within the interfacial concrete, there are no efficient methods to avoid this failure [7,8]. Thus, it is critically important to control IC debonding failure. This is especially important for slender members strengthened with thin FRP plates or sheets [9].

As shown in Figure 1, each crack that intercepts a bonded plate causes local debonding, which increases with crack width development. At the same time, interfacial stresses between the FRP plate and the concrete in the area close to the crack increase. This can cause plate debonding when the plate axial force or axial strain reaches a critical value. This failure is initiated at each crack and propagates towards adjacent cracks or the nearest plate end. Therefore, the axial force that can be developed in bonded material is limited by IC debonding resistance. The ultimate tensile strain of FRP composites is very large and much greater than the ultimate tensile strain for typical reinforcing bars. After the existing steel reinforcement yields, additional stresses are mainly carried by the FRP plate, so typically steel has yielded when IC debonding occurs [10]. After interfacial shear stress reaches the maximum shear resistance, the bond degrades, which leads to a failure like the one shown in Figure 2.

Many studies have been performed on this phenomenon, and further attempts are being made to develop new models for predicting the debonding moment in terms of PE and IC debonding failure [12,13,14,15,16]. Two experimental approaches can be distinguished in the study of debonding failure: testing small, properly prepared samples with a detailed analysis of selected parameters [13,17,18] or testing real-scale elements, which are often used to calibrate and test models [8,12,13,19,20,21]. Evaluations of debonding models have been reported by several researchers [8,13,17,19,21].

Since the EN 1992-1-1 standard revision is underway to include guidelines for strengthening existing concrete structures with carbon fiber reinforced polymers (CFRPs), the authors of the present paper examined the accuracy of selected existing IC debonding models. The models should be tested for accuracy and compared for the verification results to provide a source of guidelines for newly developed or modified standards. For this purpose, the *fib* bulletin 14 [3] third approach, Teng et al. model [22], Lu model [23], Seracino et al. model [24] and ACI 440 [25] models from design guidelines were used, as were the latest models proposed for direct implementation from the literature, namely the Said and Wu model [8] and the Elsanadedy et al. model [19]. As mentioned before, IC debonding failures are prevalent for externally bonded FRP, so it is crucial to predict them accurately when designing the strengthening of RC structures. For analysis, 58 flexural specimens with IC debonding failures of externally bonded CFRP plates or sheets were selected from the literature. The accuracy of each model was evaluated in two steps. First, the measured and predicted values of load capacity were compared. Then, the increase in experimental load capacity was compared to that estimated based on model calculations. To eliminate the influence of the safety factor, all the coefficients used were selected to obtain mean values from the guidelines for the models. Calibrating or testing models by comparing the load capacity results calculated without considering the constraints of other failure models does not accurately reflect design procedures. Therefore, unlike other analyses of this type, predicted strengths due to concrete crushing and FRP rupture were also adopted as limits. Specimens for which the failure mode was predicted incorrectly were identified.

## 2. IC Debonding Models Considered

*fib* bulletin 14:

The *fib* bulletin 14 [3] introduces three approaches, of which only the first and third approaches are suitable for design purposes. The second approach is too complicated to be applied in design; thus, it was omitted from further analysis. In the first approach, the anchorage and strain of the FRP are verified. The maximum force that can be developed in the FRP is calculated using Equation (1). This model includes the effect of the width ratio *b_f_*/*b*.
(1)$$N_{fa,\max}=\alpha c_{1}k_{c}k_{b}b\sqrt{E_{f}t_{f}f_{ctm}}$$
where the *k_b_* geometry factor is calculated from Equation (2):(2)$$k_{b}=1.06\sqrt{\frac{2-\frac{b_{f}}{b}}{1+\frac{b_{f}}{400}}}\geq 1$$
and *α* is a reduction factor typically equal to 0.9; *c*_1_ is a factor that can be calibrated, but for CFRP strips it is equal to 0.64; *k_c_* is a factor taking into account the consolidation of concrete, typically equal to 1.0; *E_f_* is the modulus of elasticity of FRP; *b_f_* is the width of FRP; *t_f_* is the thickness of FRP; *b* is the width of beam; and *f_ctm_* is the mean value of axial tensile strength of concrete.

In the third approach, the design shear force, *V_d_*, in the RC member is limited. Including some assumptions, the following conditions are given:(3)$$\varepsilon_{s1}<\varepsilon_{yd}\;\;\;\frac{V_{d}}{0.95db_{f}\left(1+\frac{A_{s1}E_{s}}{A_{f}E_{f}}\right)}\leq f_{cbd}$$
(4)$$\varepsilon_{s1}\geq \varepsilon_{yd}\;\;\frac{V_{d}}{0.95db_{f}}\leq f_{cbd}$$
where
(5)$$f_{cbd}=1.8\frac{f_{ctk}}{\gamma_{c}}$$

Teng et al. model [22]:

This model is presented in the Standards Australia design handbook [10] for calculating IC debonding resistance in beams. It includes the effect of the width ratio *b_f_*/*b*, and it is given by Equation (6).
(6)$${\left(P_{IC}\right)}_{EB}=\alpha_{EB}\beta_{p}b_{f}\sqrt{E_{f}t_{f}\sqrt{f_{cm}}}$$
where
(7)$$\beta_{p}=\sqrt{\frac{2-\frac{b_{f}}{b}}{1+\frac{b_{f}}{b}}}$$
and *α_EB_* is equal to 0.427 for the mean value; *f_cm_* is the mean value of the concrete compressive strength.

Lu model [23]:

This model also includes the effect of the width ratio *b_f_*/*b*. The equation is based on average strengths of materials. To balance safety and economy, it was modified and implemented in the Chinese concrete design code [6,26]. In this analysis, to avoid problems with overconservative results, the original Lu model was used, which is given by Equation (8):(8)$$\varepsilon_{f,IC}=\left(\frac{0.492}{\sqrt{E_{f}t_{f}}}-\frac{0.086}{L_{d}}\right)1.5\beta_{w}f_{ctm}$$
(9)$$\beta_{w}=\sqrt{\frac{2.25-\frac{b_{f}}{b}}{1.25+\frac{b_{f}}{b}}}$$
where *L_d_* is the distance from the plate end to the section where the FRP plate is fully utilized.

In comparison to the others, the Lu model is more sensitive to the impact of concrete tensile strength. Said and Wu [8] claim that the concrete strength has little effect on the IC debonding based on the calibration results of the IC debonding model proposed in their work.

Seracino et al. [24]:

This model is presented in Standards Australia design handbook [10] as a generic IC debonding resistance model. Factors used in Equation (10) were calibrated based on push–pull tests [24,27]. This model can be used to calculate IC debonding resistance not only for EB plates but also for near-surface-mounted (NSM) plates.
(10)$$P_{IC}=\alpha_{p}0.85\varphi_{f}^{0.25}f_{cm}^{0.33}\sqrt{L_{per}\left(E_{f}A_{f}\right)}<f_{f}A_{f}$$
where
$$\alpha_{p}=\{\begin{array}{c} 1.0\;\mathrm{for}\;\mathrm{mean}\\ 0.85\;\mathrm{for}\;\mathrm{lower}\;95\%\;\mathrm{confidence}\;\mathrm{limit} \end{array}$$
(11)$$\varphi_{f}=\frac{d_{p}}{b_{p}}$$
(12)$$L_{per}=2d_{p}+b_{p}$$
and *L_per_* is calculated from Equation (12) based on Figure 3.

Said and Wu model [8]:

Based on test results of 200 beams/slabs with IC debonding failures collected from the existing literature, Said and Wu calibrated and proposed this model to calculate the critical value for the FRP strain resulting in IC debonding failure, expressed by Equation (13). Based on this, an equation for debonding-moment capacity calculations was proposed with the necessary safety factors implemented. This model does not include the effect of the width ratio *b_f_*/*b*.
(13)$$\varepsilon_{deb}=\frac{0.23{\left(f_{cm}\right)}^{0.2}}{{\left(E_{f}t_{f}\right)}^{0.35}}$$

Elsanadedy et al. [19]:

This is the only model considered that was created using neural network modeling. The critical value for the FRP strain is expressed by Equation (14). This model includes the effect of the width ratio *b_f_*/*b* and is the only model to include the parameters of the tension steel reinforcement *ε_y_* and *ρ_s_*. Like the model presented above, an equation for debonding-moment capacity calculations with implemented safety factors was also proposed.
(14)$$\varepsilon_{fd}={\left(\frac{2-b_{f}/b}{1+b_{f}/b}\right)}^{0.1}{\left(\frac{\varepsilon_{y}}{nt_{f}E_{f}}\right)}^{0.4}\left(6.5+\frac{nt_{f}E_{f}}{135000}\right)\rho_{s}{}^{0.05}f_{cm}{}^{0.1}$$
where *ε_y_* is the yield strain of the steel reinforcement; *n* is the number of layers of FRP.

ACI 440.R2-17 [25]:

The approach implemented in ACI 440.2R-17 is very simple. To avoid debonding, the FRP strain is limited to the value calculated from Equation (15). The model is based on a modified version of the Teng et al. [9] model. To simplify design calculation, the effect of the width ratio *b_f_*/*b* was replaced with a typical value and subsumed into the coefficient of 0.41.
(15)$$\varepsilon_{fd}=0.41\sqrt{\frac{f_{cm}}{nE_{f}t_{f}}}\leq 0.9\varepsilon_{fu}$$

The primary input variable that appears in every presented model is the FRP axial stiffness *E_f_t_f_*. The IC debonding resistance increases with the increasing value of these parameters. The interfacial fracture energy, represented by the concrete strength, is another important parameter that should not be omitted [8]. In the *fib* bulletin 14 [3] and Lu [23] models, this parameter is represented by *f_ctm_*, while the remaining models use *f_cm_*. The power (exponent) of the strength value differentiates the influence of this parameter in individual models. As noted by Said and Wu [8], the effect of *b_f_*/*b* on debonding is quite controversial. This factor is a function of beam geometry and accounts for spreading shear stresses away from the edges of the FRP strip. The approaches implemented in the ACI 440.2R-17 [25] and Said and Wu [8] models neglect the width ratio *b_f_*/*b*. This approach is questioned by Benjamin [28], who showed that the ratio *b_f_*/*b* is important but insufficient and that the nature of the adhesive should be included in the calculation of limiting strain. Benjamin [28] points out that the Teng et al. model [22] appears to provide unconservative results for high-modulus adhesives. The Lu model [23] includes the *L_d_* variable, i.e., the distance from the plate end to the section where the FRP plate is fully utilized. Said and Wu [8] point out that this parameter’s mechanical meaning is unclear. Since bonding length is guaranteed in a correctly designed element, the value of *L_d_* cannot affect IC debonding. Elsanadedy et al. [19], based on an analysis carried out using an artificial neural network (ANN) model, determined that *nt_f_E_f_* is the most significant parameter, and *ε_sy_* is second. Typically, steel yields before IC debonding occurs; therefore, there is a rationale for introducing this parameter into the model, but this has not been confirmed by other works. The impact of individual parameters is discussed by several authors [8,19,23,28].

### 2.1. Description of the Test Database Used in the Analysis

Fifty-eight beam specimens that failed by IC debonding were selected from the existing literature. All of these RC members were strengthened with an externally bonded CFRP plate or sheet reinforcement (EBR). The experimental database was selected to include a wide range of parameters (factors describing the distribution of key parameters are shown in Table 1):Span length, *L*;Slenderness ratio, *L*/*d*;Compressive strength of concrete, *f_c_*;Yield strength of reinforcement, *f_f_*;Modulus of elasticity of steel, *E_s_*;Tensile strength of the FRP, *E_f_*;Modulus of elasticity of FRP, *f_y_*;Flexural steel reinforcement ratio, *ρ_s_* = *A_s_*_1_/*bd*, where *A_s_*_1_ is the area of steel tension reinforcement;Flexural FRP reinforcement ratio, *ρ_f_* = *A_f_*/*bd_f_*, where *A_f_* is the area of FRP tension reinforcement;Ratio of the width of the FRP sheet to the width of the concrete section, *b_f_*/*b*.

Moreover, to account for the effect of scale, RC members with different spans and slenderness were selected. Sixteen out of the fifty-eight members have a T-shaped cross section. Loading configurations vary from three- or four-point bending members to the simulation of uniformly distributed load. The type of internal steel reinforcing bars also varied (smooth bars, deformed bars). Eleven beams were subject to loading during the installation of the FRP plate.

### 2.2. Assumptions for Calculations

To determine the value of the load-carrying capacity of flexural members, the following assumptions were made:Plane sections remain plane during bending [3,25].Steel is linearly elastic, perfectly plastic [3,25].The stress–strain relationship for FRP materials is linear [3,25].The average values for steel and concrete strength were used in calculation.An equivalent rectangular stress distribution for concrete is used (Whitney stress block [25]). For debonding failure, *α*_1_—the multiplier on *f_c_* to determine the intensity of an equivalent rectangular stress distribution for concrete given by Equation (16)—and *β*_1_—the ratio of the depth of the equivalent rectangular stress block to the depth of the neutral axis—were used.For concrete crushing and rupture of FRP, bending moment capacity was calculated based on the *fib* bulletin 14 [3] approach.
(16)$$\alpha_{1}=\frac{3\varepsilon_{c}^{\prime }\varepsilon_{c}-\varepsilon_{c}^{2}}{3\beta_{1}\varepsilon_{c}^{\prime }{}^{2}}$$
where *ε*′*_c_* is the maximum strain of unconfined concrete equal to 0.002.

To reflect the procedures used in the FRP strengthening design, the predicted strength of flexural members was also limited by concrete crushing and FRP rupture failure modes. In analyses found in the literature, such an approach is not used, although it more closely reflects reality. If the IC debonding model is not the limiting failure mode, it is ignored. The predicted increase in strength, Δ*M_pred_*, was calculated as the difference between the predicted strength of the strengthened beam and the predicted strength of the beam without strengthening. The measured/experimental increase in strength, Δ*M_exp_*, was calculated as the difference between the measured strength of the strengthened beam and the measured strength of a companion control beam without strengthening.

## 3. Results

All results are reported in Table 2. Additional markings are used where the predicted failure model differs from IC debonding: CC for crushing of concrete in compression and R for rupture of FRP plate. Comparisons of the measured and predicted flexural capacity are presented graphically for each model in Figure 4, Figure 5, Figure 6, Figure 7, Figure 8, Figure 9 and Figure 10. Statistical analysis of the results is illustrated graphically in Figure 11 using box plots with values reported in Table 3.

The following criteria can be used to assess the accuracy of the models with respect to the presented results:Plots of measured versus predicted values (Figure 4, Figure 5, Figure 6, Figure 7, Figure 8, Figure 9, Figure 10, Figure 12, Figure 13, Figure 14, Figure 15, Figure 16, Figure 17 and Figure 18)—the greatest accuracy is achieved when the points are closest to the line of equality (red). Unconservative predictions are located below this line, while conservative predictions are above this line.Boxplots (Figure 11 and Figure 19)—smaller interquartile range (length of box) indicates better accuracy. Horizontal location of the box to the left indicates that the model is less conservative; a location to the right indicates that the model is more conservative.Summary statistics (Table 3 and Table 4)—mean value, median, standard deviation, maximum and minimum values, incorrect failure mode prediction—the closer to 0, the better the accuracy of the model.It is readily apparent that the *fib* bulletin 14 approach shows a wide dispersion (greatest standard deviation, 24.4%) of predictions relative to measured values (Figure 5) and a large proportion of highly overestimated results, which can be unsafe. The interquartile range is the largest. Similar problems with all *fib* bulletin 14 approaches were reported by others [8,20]. The Teng et al. and Lu models give similar results and have a slightly greater proportion of values on the conservative side. However, these two models are less accurate than the Elsanadedy et al. and Said and Wu models. The Seracino et al. model was calibrated based on push–pull tests. Such tests were meant to study the performance of the strength of the bond and transfer of the force at the FRP–concrete interface, which can also be called the shear method of testing. The results obtained in this way differ from those obtained in the tests of actual reinforced flexural elements, resulting in the Seracino et al. model being very conservative. Based on the mean value (1.1%) and median (1.6%), the best predictability was obtained from the Said and Wu model; the dispersion as measured by the standard deviation is slightly greater than several others, but the interquartile range is one of the smallest. Furthermore, 48% of the predicted values were higher than the measured capacity, including those for which crushing of concrete (CC) or rupture (R) of the FRP plate was predicted to be the failure mode. Similar behavior can be observed in the case of the Elsanadedy et al. model because IC debonding failure often occurred at almost the same load as the other failure modes. The strength values calculated for the various failure modes can be very similar. It can be assessed that the Elsanadedy et al. model takes second place after the Said and Wu model. The simple ACI 440 model is more conservative than the two mentioned above but exhibits a smaller relative error (standard deviation 12.1%). The accuracy of this model is slightly worse than that of the Teng et al. and Lu et al. models.

The second portion of the analysis compares the measured *increase* in strength (based only on experimental values) and predicted *increase* in strength (based only on calculated values). Due to the lack of results for unstrengthened control beams, the Kotynia and Kamińska [31,32] test results were not included in this comparison. From the design side, this approach checks which model results in the most accurate determination of the *increase* in the load capacity of an element after strengthening. The dispersion of results in this analysis is much greater than when only considering the total strength. This is because the relative accuracy of the predicted increase in strength is not biased by the portion of the strength attributable to the unstrengthened specimen. A graphical comparison of the values obtained for each model is shown in Figure 12, Figure 13, Figure 14, Figure 15, Figure 16, Figure 17 and Figure 18, while the values are reported in Table 2.

In Figure 12, Figure 13, Figure 14, Figure 15, Figure 16, Figure 17 and Figure 18, the results for which the model predicted a different failure mode from those observed in the tests are additionally marked. Statistical analysis of the results is presented graphically in Figure 19 with values reported in Table 4. When measuring the *increase* in strength, the predicted values closest to the measured strength increase are obtained using the ACI 440 method and the Teng et al. method. Comparing the models that fared best in the previous comparison, the Elsanadedy et al. model gives better results in predicting increased flexural strength than the Said and Wu model. This is indicated by the lesser mean value, median and standard deviation. The Seracino et al. model exhibits the smallest dispersion, but it is very conservative. For all models except those of Said and Wu and Elsanadedy et al., incorrect failure mode predictions are on the conservative side.

## 4. Conclusions

In this study, the main aim was to investigate accuracy of selected IC debonding models: *fib* Bulletin 14 [3] third approach, Teng et al. model [16], Lu model [17], Seracino et al. model [18], Said and Wu model [8], Elsanadedy et al. model [13] and ACI 440 [19]. The analysis was carried out to indicate which of the considered models is the best for newly developed standards. Several conclusions are supported by the research described in this paper:If the mean values of the deviation of the predicted load capacity from the measured value are compared, the Said and Wu (2008) model gives the best results (mean error value 1.1%), but compared to most other methods, the standard deviation is quite high. The Elsanadedy et al. model, which is one of the most complex compared to other investigated models, is next best (mean error value 3.5%). The authors of both models also provide modified versions that include appropriate safety margins. Therefore, both models are ready to be implemented in the standards and are noteworthy.The newer models (Said and Wu and Elsanadedy et al.), despite employing quite different approaches, provide very similar results and accuracy.Two models distinguished by their simplicity and ease of application—the ACI 440 and the Said and Wu model—give good results compared to other more complex models.The best match between the predicted and measured *increase* in strength was obtained using the ACI 440 method application (mean error value 1.0%).The *fib* bulletin 14 (fib 2001) approach features a wide dispersion and a large share of highly overestimated results, which can be unsafe.The largest share of incorrect failure mode predictions was observed for the Elsanadedy et al. model (52%) and the Said and Wu model (48%). This means that the values of the load capacity estimated with the use of these models exceeded those determined for flexural failures. The limitations of the calculated value of the load capacity introduced on this basis affect the results of the model accuracy analysis.Most studies presented in the literature do not give full results of materials testing. This mainly applies to FRP plate strength parameters or even test results for unreinforced beams. Therefore, the available results database is still not sufficient for the creation of a fully effective IC debonding model. More complex tests of strengthened RC beams are needed. The article indicates sources with flexural test data collected from the existing literature, which can be used in other works to test and calibrate debonding models.The analysis presented in the article was carried out with the use of models in design in mind. Therefore, a broader study of the parameters that appear in individual models has not been undertaken. The focus is on accuracy and simplicity, which are the two most important guidelines for standard development.

## Figures and Tables

**Figure 1 materials-15-07390-f001:**
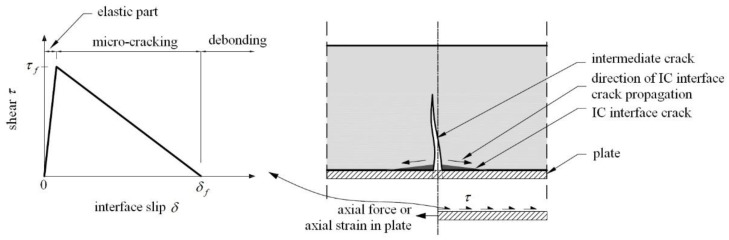
Intermediate crack (IC) [10].

**Figure 2 materials-15-07390-f002:**
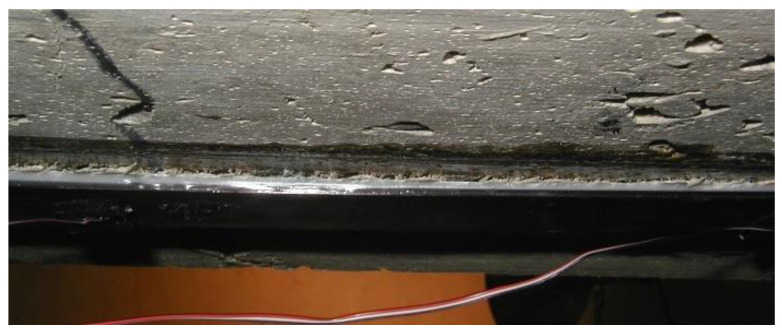
Localized debonding of FRP reinforcement in maximum moment region [11].

**Figure 3 materials-15-07390-f003:**
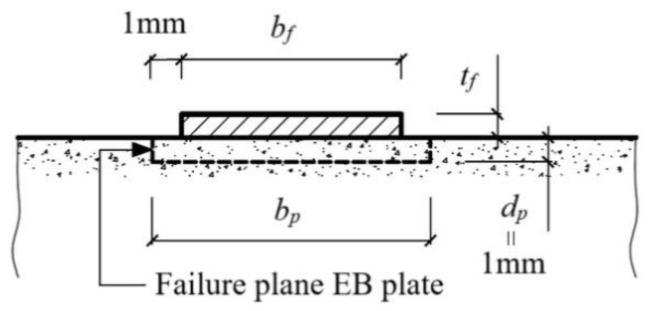
Idealized failure plane [18].

**Figure 4 materials-15-07390-f004:**
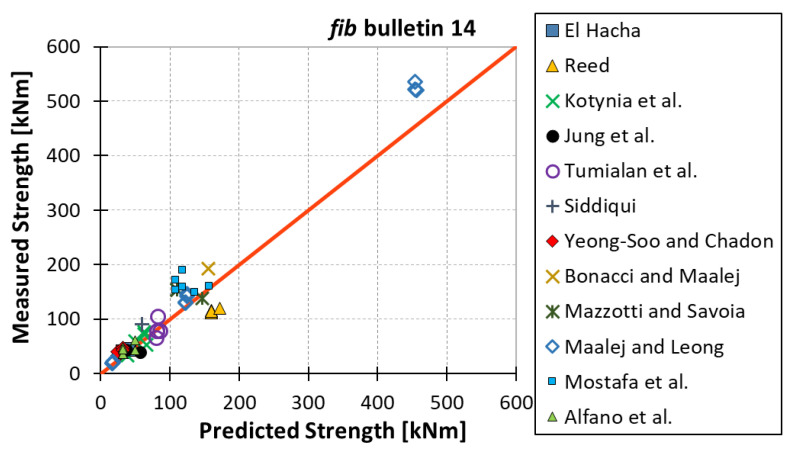
Measured flexural strength versus *fib* bulletin 14 prediction.

**Figure 5 materials-15-07390-f005:**
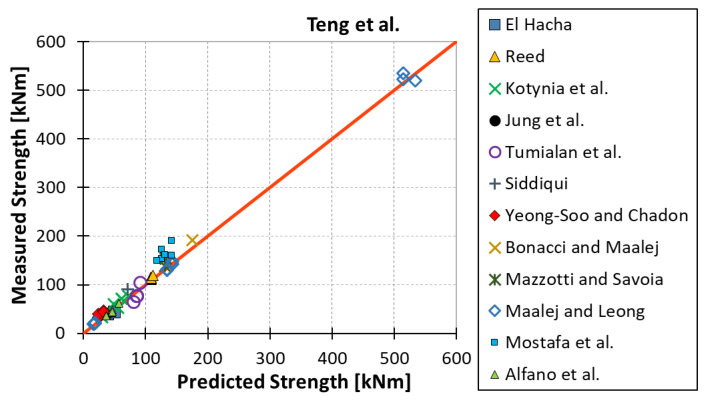
Measured flexural strength versus Teng et al. model prediction.

**Figure 6 materials-15-07390-f006:**
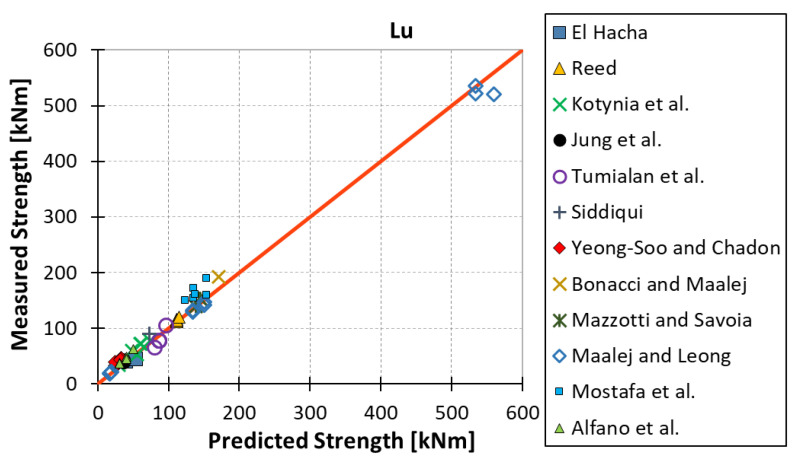
Measured flexural strength versus Lu model prediction.

**Figure 7 materials-15-07390-f007:**
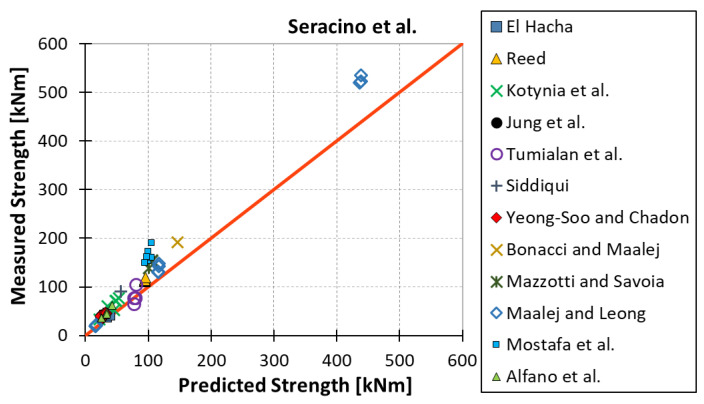
Measured flexural strength versus Seracino et al. model prediction.

**Figure 8 materials-15-07390-f008:**
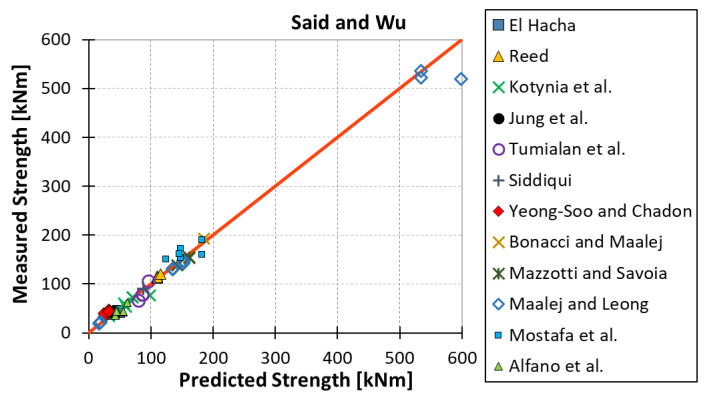
Measured flexural strength versus Said and Wu model prediction.

**Figure 9 materials-15-07390-f009:**
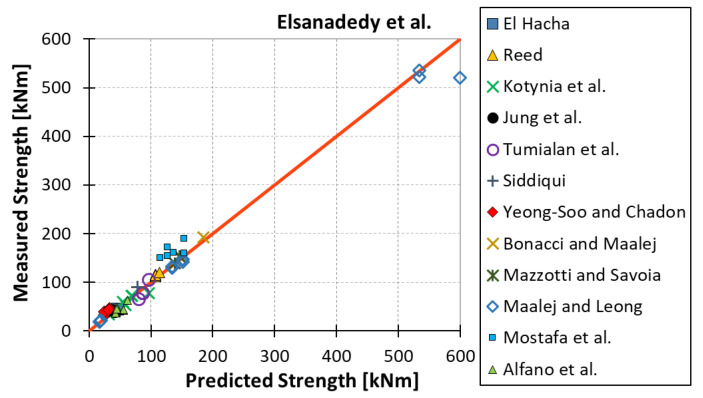
Measured flexural strength versus Elsanadedy et al. model prediction.

**Figure 10 materials-15-07390-f010:**
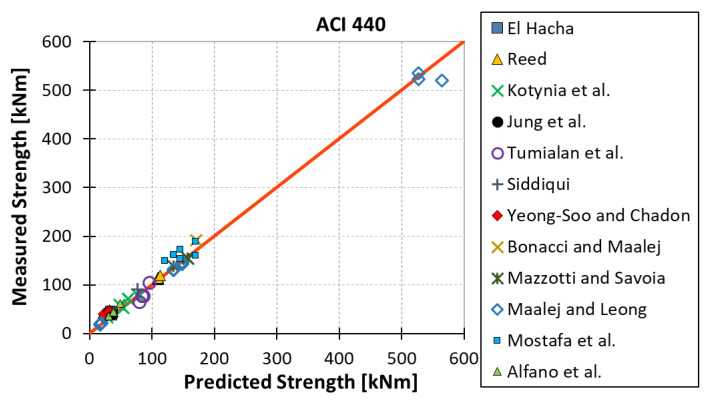
Measured flexural strength versus ACI 440 prediction.

**Figure 11 materials-15-07390-f011:**
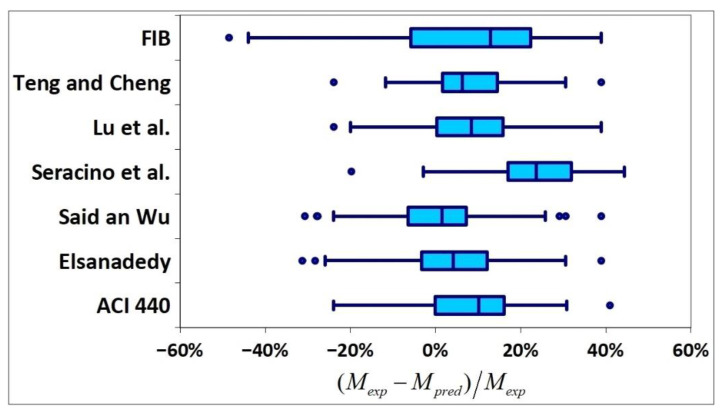
Comparison of deviation of predicted results—box plots.

**Figure 12 materials-15-07390-f012:**
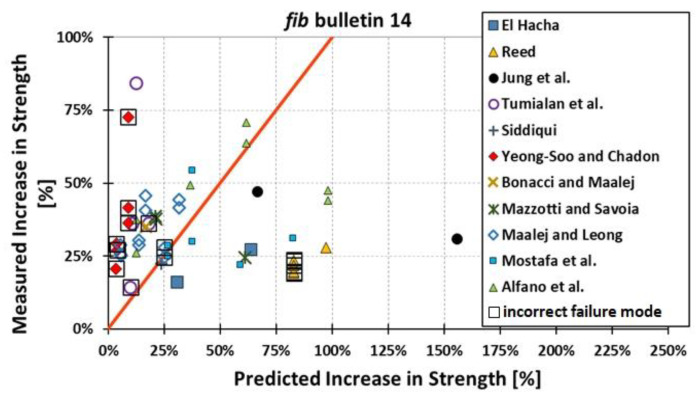
Measured increase in flexural strength versus *fib* bulletin 14 prediction.

**Figure 13 materials-15-07390-f013:**
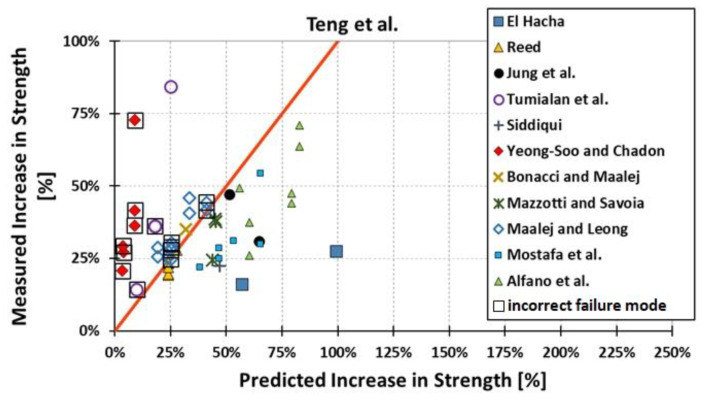
Measured increase in flexural strength versus Teng et al. model prediction.

**Figure 14 materials-15-07390-f014:**
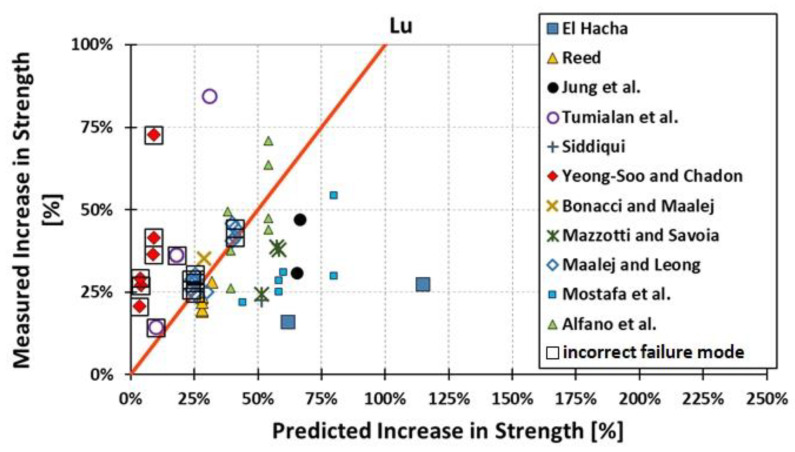
Measured increase in flexural strength versus Lu model prediction.

**Figure 15 materials-15-07390-f015:**
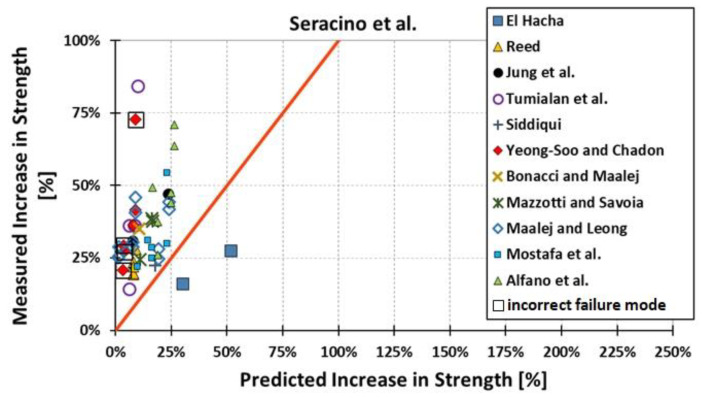
Measured increase in flexural strength versus Seracino et al. model prediction.

**Figure 16 materials-15-07390-f016:**
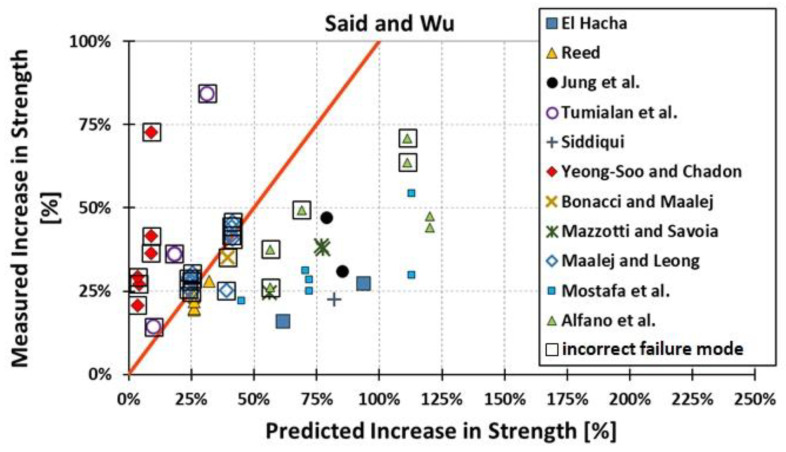
Measured increase in flexural strength versus Said and Wu model prediction.

**Figure 17 materials-15-07390-f017:**
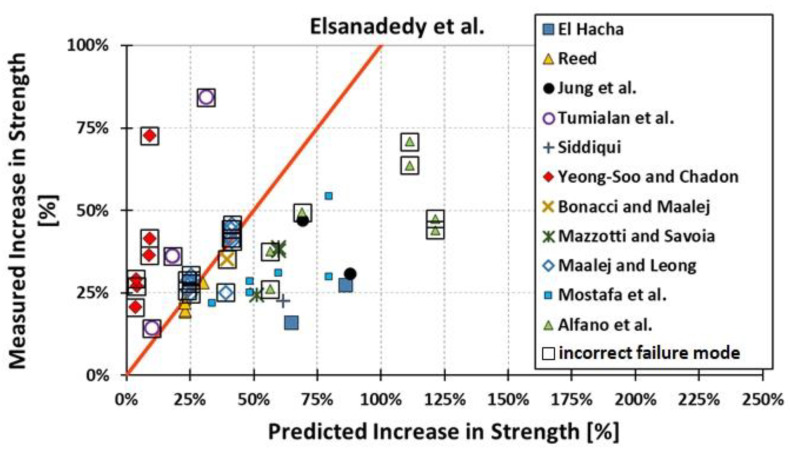
Measured increase in flexural strength versus Elsanadedy et al. model prediction.

**Figure 18 materials-15-07390-f018:**
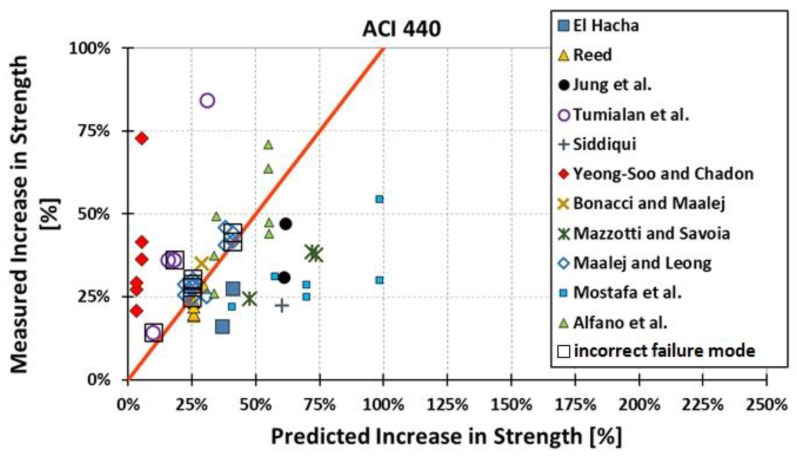
Measured increase in flexural strength versus ACI 440 prediction.

**Figure 19 materials-15-07390-f019:**
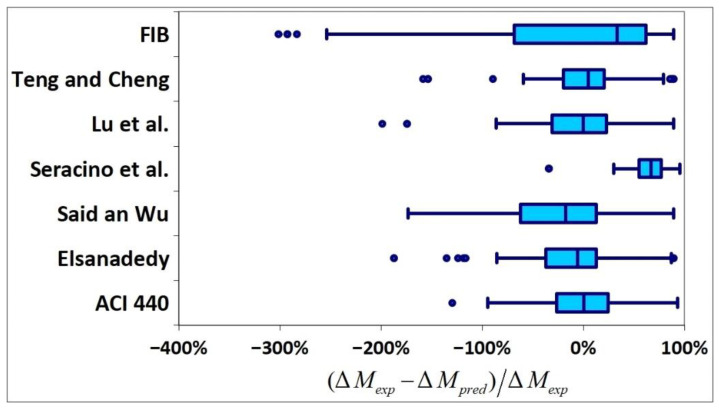
Comparison of predicted increase in flexural strength—box plots.

**Table 1 materials-15-07390-t001:** Parameters of specimens selected for analysis.

	*L*	*L*/*d*	*f_c_*	*f_f_*	*E_f_*	*f_y_*	*E_s_*	*ρ_s_*	*ρ_f_*	*b_f_*/*b*
	(mm)		(MPa)	(MPa)	(MPa)	(MPa)	(MPa)			
Mean value	3156	11.45	39.1	2852	224,000	446	197,600	0.011	0.0012	0.54
Median value	3000	11.11	42.8	2915	227,000	435	200,000	0.008	0.0011	0.45
Standard deviation	1068	2.00	13.3	821	80,600	75.4	9005	0.006	0.0009	0.36
Maximum value	4800	15.56	57.7	3900	400,000	552	207,500	0.025	0.0044	1.00
Minimum value	1500	7.55	18	846	45,000	330	180,000	0.004	0.0001	0.09

**Table 2 materials-15-07390-t002:** Parameters of specimens selected for analysis.

References	Specimen ID	Experimental		*fib*		Teng et al.		Lu		Seracino et al.		Said and Wu		Elsanadedy et al.		ACI 440	
		*Mexp*	Δ*M_exp_*	*M_pred_*	Δ*M_pred_*	*M_pred_*	Δ*M_pred_*	*M_pred_*	Δ*M_pred_*	*M_pred_*	Δ*M_pred_*	*M_pred_*	Δ*M_pred_*	*M_pred_*	Δ*M_pred_*	*M_pred_*	Δ*M_pred_*
		(kNm)	(kNm)	(kNm)	(kNm)	(kNm)	(kNm)	(kNm)	(kNm)	(kNm)	(kNm)	(kNm)	(kNm)	(kNm)	(kNm)	(kNm)	(kNm)
El Hacha and Rizkalla [29]	B2a	40.3	5.6	32.3	7.6	38.8	14.1	39.9	15.3	32.1	7.5	39.9	15.2	40.6	16.0	33.8	9.1
	B2b	40.3	5.6	32.3	7.6	38.8	14.1	39.9	15.3	32.1	7.5	39.9	15.2	40.6	16.0	33.8	9.1
	B4a	44.2	9.5	40.4	15.7	49.2	24.5	53.0	28.4	37.4	12.7	47.8	23.2	45.9	21.3	34.8	10.2
Jung et al. [30]	CPL-50-BOND	38.5	9.1	57.2	34.8	36.9	14.5	37.0	14.6	24.1	1.8	41.4	19.1	42.0	19.7	36.1	13.7
	SH-BOND	43.3	13.8	37.3	14.9	33.9	11.5	37.3	14.9	27.7	5.3	40.0	17.7	37.9	15.5	36.2	13.8
Reed et al. [10]	B1	111.7	18.0	159.7 ^R^	72.4	108.3	21.0	111.8	24.5	94.2	6.9	110.2	22.9	107.5	20.2	109.7	22.4
	B2	115.5	21.8	159.7 ^R^	72.4	108.3	21.0	111.8	24.5	94.2	6.9	110.2	22.9	107.5	20.2	109.7	22.4
	B3	112.1	18.4	159.7 ^R^	72.4	108.3	21.0	111.8	24.5	94.2	6.9	110.2	22.9	107.5	20.2	109.7	22.4
	B4	115.8	22.1	159.7 ^R^	72.4	108.3	21.0	111.8	24.5	94.2	6.9	110.2	22.9	107.5	20.2	109.7	22.4
	B5	112.1	18.4	159.7 ^R^	72.4	108.3	21.0	111.8	24.5	94.2	6.9	110.2	22.9	107.5	20.2	109.7	22.4
	B6	114.2	20.5	159.7 ^R^	72.4	108.3	21.0	111.8	24.5	94.2	6.9	110.2	22.9	107.5	20.2	109.7	22.4
	B7	119.7	26.2	172.4	85.0	111.7	24.3	115.4	27.9	95.4	8.0	115.5	28.1	113.7	26.3	113.5	26.0
Kotynia et al. [31,32]	B-08/s	72.0	-	63.2	21.5	61.5	19.8	60.4	18.8	48.1	6.4	70.4	28.8	68.8	27.1	61.4	19.8
	B0-08/s	72.0	-	63.8	22.0	62.0	20.1	61.5	19.6	48.5	6.6	71.0	29.1	69.2	27.3	62.3	20.4
	BF-06/s	59.9	-	49.7	21.3	48.2	19.7	48.3	19.9	35.4	7.0	57.0	28.5	55.1	26.6	48.1	19.7
	BF-04/0.5s	33.6	-	39.2 ^R^	20.7	30.4	11.9	30.2	11.7	22.8	4.4	32.6	14.1	31.8	13.3	28.6	10.2
	B-08S	52.8	-	66.6	25.1	56.0	14.5	55.4	13.9	46.0	4.5	59.7	18.2	59.4	17.8	54.0	12.5
	B-08M	77.0	-	63.3	21.4	72.2	30.2	74.0	32.0	53.6	11.6	98.5	56.5	97.0	55.0	78.9	36.9
Tumialan et al. [33]	A1	77.6	20.6	81.2	7.9	86.6 ^CC^	13.3	86.6 ^CC^	13.3	79.8	6.4	86.6 ^CC^	13.3	86.6 ^CC^	13.3	86.6 ^CC^	13.3
	A2	104.9	48.0	82.6	9.3	91.7	18.4	96.1	22.7	80.8	7.4	96.2 ^CC^	22.9	96.2 ^CC^	22.9	96.1	22.8
	A6	65.1	8.1	80.6 ^CC^	7.3	80.6 ^CC^	7.3	80.6 ^CC^	7.3	77.9	4.6	80.6 ^CC^	7.3	80.6 ^CC^	7.3	80.6 ^CC^	7.3
	A7	77.6	20.6	86.6 ^CC^	13.3	86.6 ^CC^	13.3	86.6 ^CC^	13.3	78.1	4.7	86.6 ^CC^	13.3	86.6 ^CC^	13.3	84.9	11.6
	C1	77.6	20.6	81.2	7.9	86.6 ^CC^	13.3	86.6 ^CC^	13.3	79.8	6.4	86.6 ^CC^	13.3	86.6 ^CC^	13.3	86.6 ^CC^	13.3
Siddiqui [34]	BFS1	90.6	16.6	59.7	11.5	70.9	22.7	73.0	24.8	56.8	8.6	87.8	39.5	78.0	29.7	77.3	29.1
Yeong-Soo and Chadon [35]	R20	31.5	8.4	24.4 ^R^	2.0	24.4 ^R^	2.0	24.4 ^R^	2.0	24.2	1.8	24.4 ^R^	2.0	24.4 ^R^	2.0	23.6	1.2
	R2L	32.7	9.6	24.4 ^R^	2.0	24.4 ^R^	2.0	24.4 ^R^	2.0	24.4	2.0	24.4 ^R^	2.0	24.4 ^R^	2.0	23.6	1.2
	R2H	39.9	16.8	24.4 ^R^	2.0	24.4 ^R^	2.0	24.4 ^R^	2.0	24.4 ^R^	2.0	24.4 ^R^	2.0	24.4 ^R^	2.0	23.6	1.2
	R30	46.0	9.8	32.6 ^CC^	1.3	32.6 ^CC^	1.3	32.6 ^CC^	1.3	32.6 ^CC^	1.3	32.6 ^CC^	1.3	32.6 ^CC^	1.3	32.4	1.0
	R3L	46.8	10.6	32.5 ^CC^	1.1	32.5 ^CC^	1.1	32.5 ^CC^	1.1	32.5 ^CC^	1.1	32.5 ^CC^	1.1	32.5 ^CC^	1.1	32.4	1.0
	R3H	43.7	7.5	32.4 ^CC^	1.1	32.4 ^CC^	1.1	32.4 ^CC^	1.1	32.4 ^CC^	1.1	32.4 ^CC^	1.1	32.4 ^CC^	1.1	32.4	1.0
Bonacci and Maalej [36]	B2	192.4	50.0	155.1	22.3	174.9	42.1	171.3	38.5	146.7	13.9	185.4 ^CC^	52.6	185.4 ^CC^	52.6	170.9	38.1
Mazzotti and Savoia [37]	TN3	138.5	27.2	146.9	55.6	131.4	40.1	138.3	47.0	101.4	10.1	142.5	51.2	138.1	46.8	134.8	43.5
	TN5	153.3	42.0	110.8	19.4	132.7	41.4	144.7	53.4	106.4	15.1	162.0	70.7	145.9	54.6	158.4	67.0
	TN8	154.4	43.1	110.5	19.3	132.3	41.0	143.7	52.5	106.1	14.9	161.4	70.2	145.7	54.4	156.9	65.7
Maalej and Leong [38]	A3	19.4	4.2	16.9 ^CC^	3.4	16.9 ^CC^	3.4	16.9 ^CC^	3.4	16.1	2.6	16.9 ^CC^	3.4	16.9 ^CC^	3.4	16.9 ^CC^	3.4
	A4	18.9	3.7	16.9 ^CC^	3.4	16.9 ^CC^	3.4	16.9 ^CC^	3.4	16.1	2.6	16.9 ^CC^	3.4	16.9 ^CC^	3.4	16.9 ^CC^	3.4
	A5	21.9	6.7	17.7	4.3	19.0 ^CC^	5.5	19.0 ^CC^	5.5	16.7	3.2	19.0 ^CC^	5.5	19.0 ^CC^	5.5	19.0 ^CC^	5.5
	A6	21.5	6.3	17.7	4.3	19.0 ^CC^	5.5	19.0 ^CC^	5.5	16.7	3.2	19.0 ^CC^	5.5	19.0 ^CC^	5.5	19.0 ^CC^	5.5
	B3	131.8	30.7	122.1	14.8	134.6 ^CC^	27.3	134.6 ^CC^	27.3	115.7	8.3	134.6 ^CC^	27.3	134.6 ^CC^	27.3	134.6 ^CC^	27.3
	B4	130.2	29.1	122.1	14.8	134.6 ^CC^	27.3	134.6 ^CC^	27.3	115.7	8.3	134.6 ^CC^	27.3	134.6 ^CC^	27.3	134.6 ^CC^	27.3
	B5	147.4	46.3	125.2	17.9	143.2	35.9	150.2	42.8	116.9	9.6	151.8 ^CC^	44.4	151.8 ^CC^	44.4	148.4	41.1
	B6	142.2	41.1	125.2	17.9	143.2	35.9	150.2	42.8	116.9	9.6	151.8 ^CC^	44.4	151.8 ^CC^	44.4	148.4	41.1
	C3	522.3	106.7	454.7	23.5	514.1	82.9	533.9 ^CC^	102.6	438.1	6.9	533.9 ^CC^	102.6	533.9 ^CC^	102.6	526.8	95.6
	C4	535.4	119.8	454.7	23.5	514.1	82.9	533.9 ^CC^	102.6	438.1	6.9	533.9 ^CC^	102.6	533.9 ^CC^	102.6	526.8	95.6
	C5	520.1	104.4	456.6	25.4	533.4	102.2	560.0	128.8	436.3	5.1	599.3 ^CC^	168.0	599.3 ^CC^	168.0	564.4	133.2
Mostafa and Razaqpur [39]	B1-F2-N	172.5	38.3	108.4	22.8	125.7	40.1	135.4	49.8	99.6	14.0	147.4	61.8	127.1	41.5	145.5	59.9
	B2-F2-N	153.8	30.8	108.4	22.8	125.7	40.1	135.4	49.8	99.6	14.0	147.4	61.8	127.1	41.5	145.5	59.9
	B1-F4-N	189.8	66.8	117.7	32.1	141.6	56.0	154.0	68.4	105.5	19.9	182.2	96.6	153.9	68.3	170.0	84.4
	B2-F4-N	159.8	36.8	117.7	32.1	141.6	56.0	154.0	68.4	105.5	19.9	182.2	96.6	153.9	68.3	170.0	84.4
	B1-F4-N b90	150.0	27.0	136.0	50.4	118.2	32.6	123.3	37.7	94.1	8.5	124.1	38.5	114.4	28.7	120.4	34.8
	B1-F8-N b90	161.3	38.3	156.2	70.5	131.3	45.7	137.1	51.4	98.1	12.4	146.1	60.5	136.8	51.2	134.8	49.2
Alfano et al. [20]	S1	42.5	13.0	50.0	24.8	45.2	20.0	38.9	13.7	31.5	6.3	55.5	30.3	55.8 ^R^	30.6	39.1	13.9
	S2	43.5	14.0	50.0	24.8	45.2	20.0	38.9	13.7	31.5	6.3	55.5	30.3	55.8 ^R^	30.6	39.1	13.9
	S3	62.0	20.5	50.0	13.4	57.0	20.4	50.5	13.9	42.6	6.0	61.9 ^CC^	25.3	61.9 ^CC^	25.3	49.2	12.6
	S5	37.6	15.6	32.6	12.4	36.9	16.7	31.0	10.9	25.5	5.3	42.6 ^CC^	22.4	42.6 ^CC^	22.4	31.2	11.1
	S6	36.0	14.0	32.6	12.4	36.9	16.7	31.0	10.9	25.5	5.3	42.6 ^CC^	22.4	42.6 ^CC^	22.4	31.2	11.1
	S7	48.4	13.2	32.6	3.6	46.4	17.4	40.3	11.4	34.4	5.5	45.3 ^CC^	16.3	45.3 ^CC^	16.3	38.7	9.7
	S8	44.4	9.2	32.6	3.6	46.4	17.4	40.3	11.4	34.4	5.5	45.3 ^CC^	16.3	45.3 ^CC^	16.3	38.7	9.7

^CC^ predicted failure mode—crushing of concrete in compression. ^R^ predicted failure mode—rupture of FRP plate.

**Table 3 materials-15-07390-t003:** Comparison of deviation of predicted results.

(*M_exp_* − *M_pred_*)/*M_exp_* (%)	*fib*	Teng et al.	Lu et al.	Seracino et al.	Said and Wu	Elsanadedy et al.	ACI 440
Mean value	6.0%	8.1%	7.7%	23.7%	1.1%	3.5%	8.7%
Median	12.9%	6.4%	8.5%	23.6%	1.6%	4.1%	10.2%
Maximum value	38.9%	38.9%	38.9%	44.4%	38.9%	38.9%	41.0%
Minimum value	−48.5%	−23.9%	−23.9%	−19.8%	−30.7%	−31.3%	−23.9%
Standard deviation	24.4%	12.3%	12.3%	12.0%	14.2%	14.8%	12.1%
Incorrect failure mode prediction	29%	28%	31%	7%	48%	52%	16%

**Table 4 materials-15-07390-t004:** Comparison of deviation of predicted results.

(Δ*M_exp_* − Δ*M_pred_*)/Δ*M_exp_* (%)	*fib*	Teng and Cheng	Lu et al.	Seracino et al.	Said an Wu	Elsanadedy et al.	ACI 440
Mean value	−16.6%	−2.0%	−6.7%	61.3%	−25.7%	−15.4%	1.0%
Median	33.5%	4.8%	−0.1%	66.9%	−17.5%	−5.8%	0.5%
Maximum value	89.3%	89.3%	89.3%	95.2%	89.3%	89.3%	92.9%
Minimum value	−301.5%	−158.6%	−198.9%	−34.3%	−173.4%	−187.3%	−129.7%
Standard deviation	118.2%	54.8%	60.2%	27.9%	69.0%	64.1%	48.6%
Incorrect failure mode prediction	29%	28%	31%	7%	48%	52%	16%

## Data Availability

Not applicable.

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
