# Peer review of "Intermediate Crack Debonding of Externally Bonded FRP Reinforcement—Comparison of Methods"

_materials, 2022, doi:10.3390/ma15207390_

Round 1
Reviewer 1 Report
I have reviewed the manuscript "Intermediate-Crack Debonding of Externally Bonded FRP Reinforcement-Comparison of Methods" by Pawel Tworzewski, Jeffrey K. Alexy and Robert W. Barnes. This paper presents comparison of seven selected intermediate-crack (IC) debonding models by calculating 58 flexural specimens from the existing literature. In general, the topic is of interest, however, the paper has significant flaws. The conclusion presented in the paper is relatively superficial and it is lack of innovation.
Detailed comments are presented as follow:
(1) The paper evaluated seven existing IC debonding models by using specimens from the existing literature. Basically, it is just a summarize of the existing studies and lack of innovation.
(2) The results presented in Section 3 are relatively superficial. This paper only presents the comparison of the prediction accuracy of these seven models. However, the difference in mechanism of these seven modeling is not analyzed. In other word, the reason why these models present different prediction accuracy should be further discussed.
(3) Fig 1 is not clear and meaningless. Maybe it can be removed.
(4) In Section 2, it is suggested to add the meaning of the parameters (such as Ef and Af) in the models considered. This makes it easier to follow.
(5) Please check the format of the parameters, such as Line 87: Where the kb geometry factor…
Author Response
|
(1) The paper evaluated seven existing IC debonding models by using specimens from the existing literature. Basically, it is just a summarize of the existing studies and lack of innovation. |
Description in the text has been added - Line 109-111: Unlike other analyses of this type, the predicted strength of flexural members was limited due to concrete crushing and FRP rupture. It was also indicated for which samples the failure mode was mispredicted. |
|
(2) The results presented in Section 3 are relatively superficial. This paper only presents the comparison of the prediction accuracy of these seven models. However, the difference in mechanism of these seven modeling is not analyzed. In other word, the reason why these models present different prediction accuracy should be further discussed. |
Description of the parameters in the models and their meaning has been added - Line 191-213;
Description in the text has been added - Line 441-444: The analysis presented in the article was carried out with the use of models in design in mind. Therefore, a broader study of the parameters that appear in individual models has not been undertaken. The focus is on accuracy and simplicity, which are the two most important guidelines for standard development. |
|
(3) Fig 1 is not clear and meaningless. Maybe it can be removed. |
removed |
|
(4) In Section 2, it is suggested to add the meaning of the parameters (such as Ef and Af) in the models considered. This makes it easier to follow. |
Description in the text has been added |
|
(5) Please check the format of the parameters, such as Line 87: Where the kb geometry factor… |
corrected |

Reviewer 2 Report
This paper aims to build an experimental database of IC debonding failure, and further compares the calculation results obtained by different IC debonding models with the experimental results. The authors made an effort to tell the most effective method in evaluating the IC debonding failure. The work is valuable but lacks rigorous logic. The authors should consider the comments and answer the questions as follows:
1. What kind of FRP materials do the authors want to discuss in the paper? As different FRP materials present different mechanical behaviors, for example, CFRP and BFRP, it’s hard to simply use the word “FRP” without distinction.
2. In line 87, kb should be wrongly expressed.
3. Necessary symbols, e.g., fc and fy, should be added in lines 147-148.
4. As "EBR" appears few in this paper, it’s unsuitable to be used as a keyword.
5. More specific information about seven IC debonding models should be given in the introduction. In addition, the authors should give more comments on the significance and innovation of the research work in this part.
6. Please revise the format of Table 4. Additionally, there’re some grammar and spelling errors throughout the text, the authors should check and revise the paper more carefully.
7. As different correction coefficients are usually adopted in different models, it’s important to distinguish the difference and evaluate their effect on the experimental results when the experimental database was built. It seems that the authors have neglected this issue.
8. It’s necessary to provide the criterion to judge the IC debond failure.
9. In the conclusion, the authors should explain the significance and shortcomings of the research work, instead of repeating the results obtained before.
Author Response
|
What kind of FRP materials do the authors want to discuss in the paper? As different FRP materials present different mechanical behaviors, for example, CFRP and BFRP, it’s hard to simply use the word “FRP” without distinction. |
Description in the text has been added - Line 16-18, 101-102, 216-217; |
|
In line 87, kb should be wrongly expressed. |
corrected |
|
Necessary symbols, e.g., fc and fy, should be added in lines 147-148. |
Description in the text has been added |
|
As "EBR" appears few in this paper, it’s unsuitable to be used as a keyword. |
Description in the text has been added - Line 16-18, 101-102, 216-217; |
|
More specific information about seven IC debonding models should be given in the introduction. In addition, the authors should give more comments on the significance and innovation of the research work in this part. |
Description of the parameters in the models and their meaning has been added - Line 191-213; Description in the text has been added or modified |
|
Please revise the format of Table 4. Additionally, there’re some grammar and spelling errors throughout the text, the authors should check and revise the paper more carefully. |
corrected |
|
As different correction coefficients are usually adopted in different models, it’s important to distinguish the difference and evaluate their effect on the experimental results when the experimental database was built. It seems that the authors have neglected this issue. |
Description in the text has been added - Line 106-107: To eliminate the influence of the safety factor, all the coefficients used were selected to obtain mean values from the guidelines for the models. |
|
It’s necessary to provide the criterion to judge the IC debond failure. |
Description in the text has been added - Line 289-301 |
|
In the conclusion, the authors should explain the significance and shortcomings of the research work, instead of repeating the results obtained before. |
corrected |

Reviewer 3 Report
Intermediate-Crack Debonding of Externally Bonded FRP Reinforcement Comparison of Methods. The article is interesting. A few observations are given below,
The abstract is not clear. Objectives and aims are missing in the abstract. Abstract need revision with some quantitative results.
Figure 1 is not clear.
Some more latest studies are required in the introduction section to further highlight the importance of this study
Equations must be quoted with proper references
the explanation of "fib bulletin 14" is not clear to understand.
Author should include a general description of "IC Debonding Models Considered" (section 2) before the specific model explanations.
Quoted with proper references, based on what references and theorems section "2.2. Assumptions for calculations" assumptions has been made.
Conclusions are too limited to proof the significant outcome of this study.
Author Response
|
The abstract is not clear. Objectives and aims are missing in the abstract. Abstract need revision with some quantitative results. |
modified |
|
Figure 1 is not clear. |
removed |
|
Some more latest studies are required in the introduction section to further highlight the importance of this study |
added |
|
Equations must be quoted with proper references |
corrected |
|
the explanation of "fib bulletin 14" is not clear to understand. |
modified |
|
Author should include a general description of "IC Debonding Models Considered" (section 2) before the specific model explanations. |
Description of the parameters in the models and their meaning has been added - Line 191-213 |
|
Quoted with proper references, based on what references and theorems section "2.2. Assumptions for calculations" assumptions has been made. |
corrected |
|
Conclusions are too limited to proof the significant outcome of this study. |
modified |

Round 2
Reviewer 1 Report
This manuscript is of good quality and is recommended to be accepted.
Author Response
Thank you for your valuable comments.
Manuscript was checked by a co-author from U.S. who is fluent in English writing and English improvements were introduced in the resubmitted version

Reviewer 2 Report
I have no further comments.
Author Response

(The authors gave the same response as above.)
